# Estimation of the Annual Effective Dose Due to the Ingestion of ^210^Pb and ^210^Po in Crops from a Site of Coal Mining and Processing in Southwest China

**DOI:** 10.3390/molecules27072112

**Published:** 2022-03-25

**Authors:** Chenxiao Wang, Qifan Wu, Ziqiang Pan, Senlin Liu, Zhonggang Cao, Yilin Yu

**Affiliations:** 1Department of Engineering Physics, Tsinghua University, Beijing 100084, China; 2China Institute of Atomic Energy, Beijing 102413, China; 3Science and Technology Commission, CNNC—China National Nuclear Corporation, Beijing 100822, China; anzq@cnnc.com.cn (Z.P.); slliu@ciae.ac.cn (S.L.); 4Zhejiang Province Environmental Radiation Monitoring Center, Hangzhou 310012, China; cazgg2005@aliyun.com; 5Yunnan Provincial Radiation Environment Supervision Station, Kunming 650034, China; ynhb@163.com

**Keywords:** natural radioactivity, risk assessment, ^210^Pb and ^210^Po, radiological impact, polluted mine site

## Abstract

The exploitation of mineral resources may cause the environmental release of radionuclides and their introduction in the human trophic chain, affecting public health in the short and long term. A case study of the environmental radiation impact from coal mining and germanium processing was carried out in southwest China. The coal mines contain germanium and uranium and have been exploited for more than 40 years. The farmlands around the site of the coal mining and germanium processing have been contaminated by the solid waste and mine water to some extent since then. Samples of crops were collected from contaminated farmlands in the research area. The research area covers a radius of 5 km, in which there are two coal mines. ^210^Pb and ^210^Po were analyzed as the key radionuclides during the monitoring program. The average activity concentrations of ^210^Pb and ^210^Po in the crops were 1.38 and 1.32 Bq/kg in cereals, 4.07 and 2.19 Bq/kg in leafy vegetables and 1.63 and 1.32 Bq/kg in root vegetables. The annual effective doses due to the ingestion of ^210^Pb and ^210^Po in consumed crops were estimated for adult residents living in the research area. The average annual effective dose was 0.336 mSv/a, the minimum was 0.171 mSv/a and the maximum was 0.948 mSv/a. The results show that the crops grown on contaminated farmland contained an enhanced level of radioactivity concentration. The ingestion doses of local residents in the research area were significantly higher than the average level of 0.112 mSv/a in China, and the world average level of 0.042 mSv/a through ^210^Pb and ^210^Po in crop intake, respectively.

## 1. Introduction

Naturally occurring radioactive material (NORM) comprises radionuclides associated with the ^238^U and ^232^Th decay chains as well as ^40^K [1]. These radionuclides are present in many natural resources. During mining and processing, radionuclides in fossil fuels transfer into the waste, which is usually disposed of in spoil heaps and tailing ponds on the mining site [2]. In some mining enterprises, proper management and adequate controls are not taken to minimize the impact on the environment [3,4]. The radionuclides are released into the air, soil and surface water from the mine waste as a result, and then public health is affected [5].

China is the biggest producer of coal but only a few mines are associated or paragenetic with uranium [6,7,8,9,10]. These coal mines also reserve various resources such as aluminum, iron, silicon, germanium and gallium [11,12,13,14]. Some coal mines associated with uranium or other metals cause radioactive impact on the surrounding areas by several pathways, such as the emission of dust (gas) during the mining and smelting process, the accumulation of coal slag and the discharge of liquid effluents and residues from germanium purification [15,16,17,18,19,20,21].

Specifically, the technological process of germanium production consists of a thermal process and a chemical process. In the thermal process, germanium, uranium and other natural radionuclides in the raw coals are concentrated in fly ash when raw coals are burning in the furnace. The germanium content of the fly ash removed from the bag filters is 2.32% compared with 0.053% in the raw coals, while radioactivity concentrations in the coal became significant when concentrated in the ash. Germanium dioxide is produced when the fly ashes are then treated by chlorination oxidation in the chemical process. The environmental radiological impact in this research area mainly results from the release of radioactive substances during the mining and producing process, as well as the waste residue left over from the past.

After the radioactive waste from mineral exploitation is released into the environment, the radionuclides are transferred and redistributed in different types of receptor environments [4]. In cropland, which is one of the receptors, exposure pathways associated with the cultivation of agricultural plants can be assessed. Radionuclide accumulation in plants is determined based on radionuclide concentrations in the topsoil layer and it accounts for the root transfer from the contaminated soil and the interception by plant leaves [22,23]. Residents living around the mines consume foodstuffs cultivated in the contaminated croplands and as a result, the potential exposure risks of ingesting radionuclides for local residents increase significantly. Therefore, our study aims to (1) investigate the radioactivity level of the crops cultivated in local croplands, (2) estimate the ingestion doses of adult residents living in the research area and (3) assess the environmental radiological impact caused by coal mines located in southwest China. An area with a radius of 5 km was defined as the research area with two coal mines located in this area. Cereals, root vegetables and leafy vegetables planted in the research area were sampled. They are the most common crop types in the local diet. ^210^Pb and ^210^Po in crops were selected as the key radionuclides that contribute the most to ingestion doses among the natural radionuclides in ^238^U and ^232^Th decay chains [1]. 

## 2. Materials and Methods 

### 2.1. Site Description

The research area is in southwestern China, as shown in Figure 1. In the research area, coal mines contain germanium and uranium. They have only been mined for local livelihoods and industry combustion since the beginning. Since the 1970s, the coal mines were exploited for germanium production [2]. The average activity concentration of the natural radionuclide ^238^U in the coal ores is 624 Bq/kg, while the maximum is 2.17 kBq/kg [3]. The research area is located in the countryside with a population density of 195 people per square kilometer. The climate is mild in this area and is suitable for planting common cereals and vegetables. 

The monitoring program was carried out in 21 sampling locations. The ingestion doses were estimated for adult residents from 11 villages. The population of the villages ranges from 206 to 4367 people. The locations of the sampling sites, coal mines and villages are shown in Figure 2.

The research area covers a 5 km radius circle centered on the thermal coal processing plant and 2 coal mines are located there. One of the two coal mines is Mine XY which is located in the western part of the area and is 2.5 km from the center, while the other mine is Mine CJ which is located in the northeast part of the area and is about 5 km far away from the center of this area. The annual production of germanium for Mine XY is 50,000 t, while the annual coal production of Mine CJ is about 8000 to 10,000 t.

There are 4 villages located in the area surrounding Mine CJ, which are Village MW, Village DG, Village LD and Village JZ. The farmlands in the area are distributed along a small river. Village MW is nearest to Mine CJ and is upstream, while the other villages are downstream of the river from Mine CJ. There are 4 villages near Mine XY, which are Village DQP, Village SCK, Village MCD and Village AK. The villages and farmlands are located in a flat area along the valley. 

In the research area, Village CHB and Village DTH are far from the two coal mines. Village CHB is about 6 km from Mine XY, but the farmland around the village is down-stream in the valley and may be affected by the river release from the mine. Village DTH is located at a higher altitude than Mine XY, but the farmland around this village is exposed to large quantities of mining waste and ore tailings.

Village MTDZ is outside the research area and on a different side of the mountain. Crops growing in the farmland around the village are not affected by mining and processing. Therefore, concentrations of radionuclides in the samples from Village MTDZ were considered as a reference level, or the background level of this area. 

### 2.2. Analysis Methods

Samples of corn, wheat, rice, greens and plantains grown in the research area were collected for radionuclides analysis. In the laboratory, the samples were washed and dried. The ratios of dry weight to fresh weight for cereals, root vegetables and leafy vegetables were 1.0, 0.3 and 0.1, respectively. Then the dry samples were pretreated to analytical samples. 

To measure the activity concentrations of ^210^Pb, the crop samples were solubilized by chemical digestion at first in the laboratory. A stable Pb carrier was added to the solubilized samples for the estimation of the ^210^Pb recovery rate during the chemical/physical processes. ^210^Pb and Pb carriers were adsorbed and concentrated by Fe (OH)_3_ co-precipitation, underwent heating and were isolated from most interferences by an anion exchange resin. The PbSO_4_ precipitate was purified by repeating the precipitation process three times and then it was stored at environment temperature for one month. The activity concentration of ^210^Pb was estimated through the beta count of daughter ^210^Bi [24].

The activity concentrations of ^210^Po were measured through its 5.30 MeV alpha particle emission. ^209^Po (4.88 MeV, t_1/2_ = 109 y) was used as the internal tracer. The polonium tracer was added at first, then samples were dissolved by concentrated 20 mL HNO_3_, respectively. After the addition of 2 mL HCl, the solutions were evaporated to dryness. Ascorbic acid was added to reduce Fe^3+^ to Fe^2+^ and then polonium was plated on a silver disc from a diluted HCl medium. A PIPS–alpha spectrometer (PIPS: Passivated Implanted Planar Silicon) with 4 detectors ranging from 450 mm^2^ to 1200 mm^2^ was used to measure the alpha activities. The detection limit of ^210^Po in the samples was 0.0003 Bq. Sample count times were 864,000 s and the maximum counting errors were in the order of ± 10% [25,26].

### 2.3. Dose Calculations

For public exposure to radionuclides, ingestion is a significant route of intake. Elements incorporated into food may be more readily absorbed from the gastrointestinal (GI) tract than inorganic forms of these elements, according to ICRP Publication 103 [27].

The ingestion doses for adults are calculated by the following general equation:E_ing,p_ = C_p,i_ × DF_ing_ × H_p_(1)
where E_ing,p_ is the annual effective dose from the consumption of nuclide i in foodstuff p (Sv/a), C_p,i_ is the concentration of radionuclide i in foodstuff p at the time of consumption (Bq/kg), DF_ing_ is the dose coefficient for the ingestion of radionuclide i (Sv/Bq) and H_p_ is the consumption rate for foodstuff p (kg/a).

Dose coefficients provide numerical links between E_ing_ and measurable quantities, which means the intake of radionuclides by ingestion in this paper. The dose coefficients used for the ingestion of ^210^Pb and ^210^Po are shown in Table 1 [28].

## 3. Results

### 3.1. Monitoring Data

Samples of raw materials, products and waste residues were collected and analyzed, as shown in Table 2, Table 3 and Table 4. The radioactivity concentrations of raw materials and residues exceeded 1Bq/g, while the radioactivity concentrations of ^210^Pb and ^210^Po in intermediate products or fly ashes and chemical residues reached an order of magnitude of 10^4^ Bq/kg.

In Table 2, samples of coal ores were collected from the locations MW, DZ, HJ, MT and JCB and the samples of the residues were collected from these coal ores after thermal processing.

In Table 3, the samples of fly ashes were mixed samples. The samples of the chemical residues included acidic residues, alkaline residues and neutralized residues.

Radioactivity concentrations in the water collected in the research area are shown in Table 4. Although the wastewater was treated, the radioactivity level in the discharged water was still higher than the background level. Generally, the nearby surface water was potentially polluted by wastewater discharge to the river.

The activity concentrations of ^210^Pb and ^210^Po in the samples were analyzed. Samples of cereals included corn, wheat and rice. Leafy vegetables were greens. Root vegetables were plantains. The values of the average and range of activity concentrations analyzed in each kind of crop are shown in Table 5.

### 3.2. Ingestion Dose Estimation

The dose ingestion of radionuclides from the local populations was estimated considering the activity concentrations of different foodstuffs and the investigated food habits of adults [29]. The adults’ consumption rate of cereals, root vegetables and leafy vegetables are shown in Table 6.

The ingestion doses of adult villagers living in the research area and Village MTDZ are shown as follows. The contributors of different types of crops and the contributors of different radionuclides are shown in Table 7. As we can see, the maximum contributor of the ingestion doses was from cereals.

## 4. Discussion

### 4.1. The Correlation of the Radionuclides in Crops and Soils

Crop and soil samples were paired according to their sampling locations. The concentrations of ^210^Pb and ^210^Po in the corn and soil samples are shown in Figure 3. Due to the near equilibrium in the decay chains, the concentrations of ^210^Pb and ^210^Po in soil were almost equal.

In Figure 3, the correlation between the activity concentrations of ^210^Pb and ^210^Po in the corns and soils is not obvious. This situation indicates that the transfer from soil to crop may be affected by various factors, such as the presence of other contaminants, differences in soil characteristics and the chemical composition of irrigation water, among others.

The biological aspects concerned with mineral uptake also control the soil-to-plant transfer of radionuclides, imposing changes in ecological parameters such as transfer factors and the translocation factor from the roots to the aerial parts. The different accumulation of radionuclides in crops can be seen by comparing the activity concentrations in different types of crops collected in the same position. In Village DTH, two corn samples and two rice samples were collected and in Village MTDZ, one corn sample and two leafy vegetable samples were collected, while in Village DQP, three root vegetable samples and one leafy vegetable sample were collected. The chart in Figure 4 shows the data of the activity concentrations for ^210^Pb and ^210^Po in several types of consumable plants for three villages (DTH, DQP and MTDZ) and the differences can be acknowledged.

In Village DTH, the activity concentrations of ^210^Pb or ^210^Po in rice were much higher than that in corn. In spite of the plants being under the same soil conditions, the transfer factors for ^210^Pb/^210^Po were clearly higher for rice than for corn. This situation is the same as the description in the Technical Reports Series No. 472 [30] published by the IAEA, in which the transfer factors of ^210^Pb or ^210^Po in soil to rice are 8.4 × 10^−3^ and 1.3 ×10^−2^ and the transfer factors of ^210^Pb or ^210^Po in soil to corn are 8.5 × 10^−4^ and 2.4 × 10^−4^, respectively.

In Village MTDZ, the activity concentrations of ^210^Pb or ^210^Po in leafy vegetables were higher than that in corn. The concentration ratios of ^210^Pb or ^210^Po for leafy vegetables were 8.2 × 10^−2^ and 7.4 × 10^−3^ in soil, respectively, which were higher than that for corn.

In Village DQP, the activity concentrations of ^210^Pb or ^210^Po in leafy vegetables were much higher than root vegetables.

### 4.2. Ingestion doses

The references levels of intake doses ^210^Pb / ^210^Po for China and the rest of the world due to the consumption of cereals, leafy vegetables and root vegetables were estimated, assuming the same consumption rates derived for the research area in this work. The reference levels of ^210^Pb and ^210^Po in cereals, root vegetables and leafy vegetables in China and the world are shown in Table 8 [1,23].

Concentrations of naturally occurring radionuclides in foods vary widely with different background levels and contaminated soil, climate and agricultural conditions [31]. Meanwhile, the traditional Chinese diet is quite different from western countries, which may lead to different ingestion doses.

The average annual effective dose due to the ingestion of ^210^Pb and ^210^Po in crops for adults living in the research area was 0.336 mSv/a, with the range from 0.171 to 0.948 mSv/a. The doses for adults living in Village MTDZ was 0.125 mSv/a, as a reference level, or the background level of the research area. The background level was higher than the reference level for both China and the world. The estimated ingestion doses of adults living in the villages around the coal mines are shown in Figure 5.

The ingestion dose for adults living in Village MW was 0.697 mSv/a. Village MW is close to Mine CJ. There is a river passing through Village MW to Village DG, Village LD and Village JZ. There is a source for irrigation in this area and radionuclides are released downstream from Mine CJ, potentially affecting agricultural lands. The study indicated that the ingestion doses of adults living in Village DG, Village LD and Village JZ were lower than in Village MW, but higher than the reference level. This implies that the radiological impact of mining activity may decrease with distance. Although Village DQP is close to Mine XY, the ingestion dose of adults living in Village DQP was 0.213 mSv/a. Village DQP is less affected because it is located upstream of Mine XY.

The ingestion dose of adults living in Village DTH was 0.948 mSv/a, which was the highest in the research area. This implies that farmlands in Village DTH are contaminated more seriously than others. The estimated ingestion dose for residents living in Village MTDZ was 0.125 mSv/a, which was the lowest and similar to the background dose exposure from ingestion in this area. It is close to the China reference level of 0.112 mSv/a. Village MTDZ was less polluted by the exploitation of coal mines.

## 5. Conclusions

In our study, twenty-one samples of cereals, root vegetables and leafy vegetables grown on the farmlands in the research area were collected. The activity concentrations of key radionuclides of ^210^Pb and ^210^Po in the samples were analyzed. The radioactivity concentrations in rice were the highest, while that in corn were the lowest. The average activity concentration of ^210^Pb for rice, wheat, corn, greens and plantains was 7.26 Bq/kg, 1.92 Bq/kg, 0.35 Bq/kg, 4.07 Bq/kg and 1.63 Bq/kg, and the value of ^210^Po for rice, wheat, corn, greens and plantains was 5.81 Bq/kg, 2.73 Bq/kg, 0.45 Bq/kg, 2.19 Bq/kg and 1.32 Bq/kg, respectively. According to China’s National Standard of the Limited Concentrations of Radioactive Materials in Foods [32], the limited concentration of ^210^Po for cereals, root vegetables and leafy vegetables is 6.40 Bq/kg, 2.80 Bq/kg and 5.30 Bq/kg, respectively. Therefore, the activity concentration of ^210^Po in all crops did not exceed the limited concentrations of China’s national standard.

The effective doses due to the ingestion of ^210^Pb and ^210^Po in crops were estimated for adults living in 10 villages in the research area, while the result in Village MTDZ was considered as the reference level. The average ingestion dose of adults living in the research area was 0.336 mSv/a, with a range from 0.171 to 0.948 mSv/a. The estimated result for MTDZ’s villagers was 0.125 mSv/a. Compared with the reference levels of China and the world, 0.112 and 0.042 mSv/a, respectively, the estimation values for local residents were much higher. The ingestion doses of local residents were about three times higher than the China reference level and eight times higher than the world reference level. The results indicated that the mining and processing of coal mines for 40 years in the research area has resulted in an obvious radiological impact on the environment and has led to a significant increase in the ingestion doses of local residents. The results also revealed that when the farmlands in the villages were close to coal mines and located downstream of mining activity, the ingestion doses of adults living in villages were higher. In order to reduce potential exposure risks, it is suggested that crops with low concentration ratios should be cultivated in farmlands that are slightly contaminated. For farmlands that are seriously polluted, arrangements such as soil remediation are required.

## Figures and Tables

**Figure 1 molecules-27-02112-f001:**
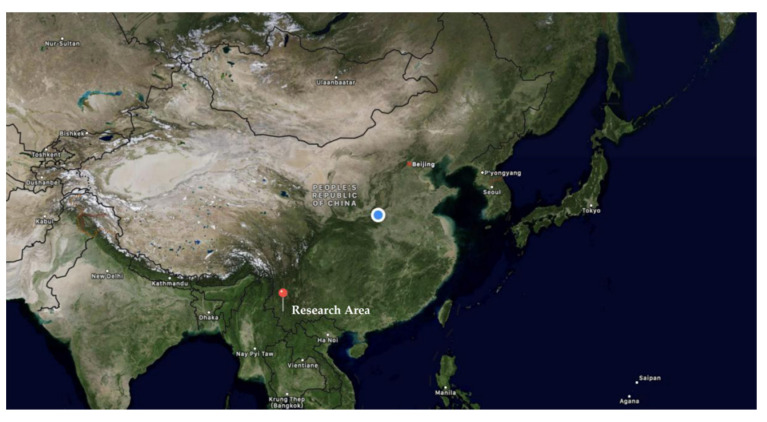
The location of the research area.

**Figure 2 molecules-27-02112-f002:**
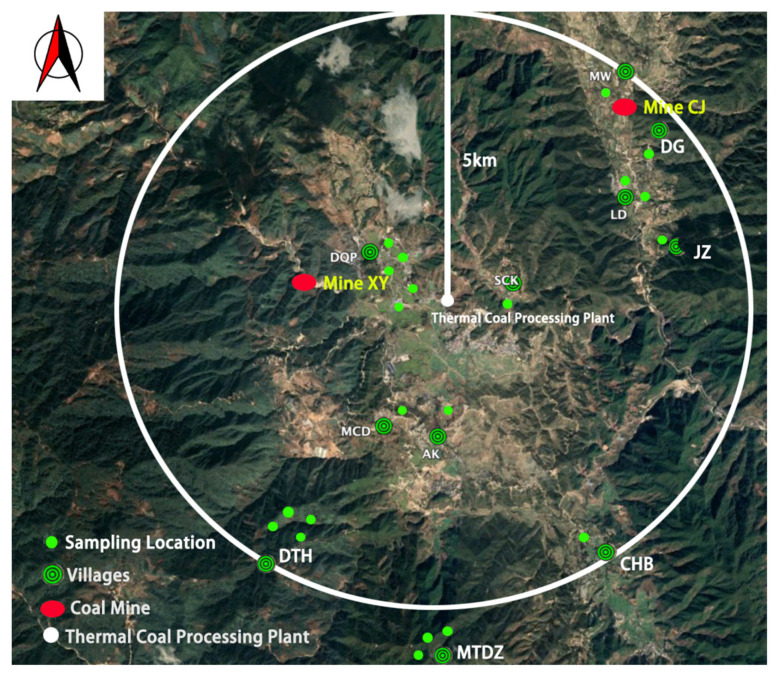
The locations of the sampling sites, coal mines and villages.

**Figure 3 molecules-27-02112-f003:**
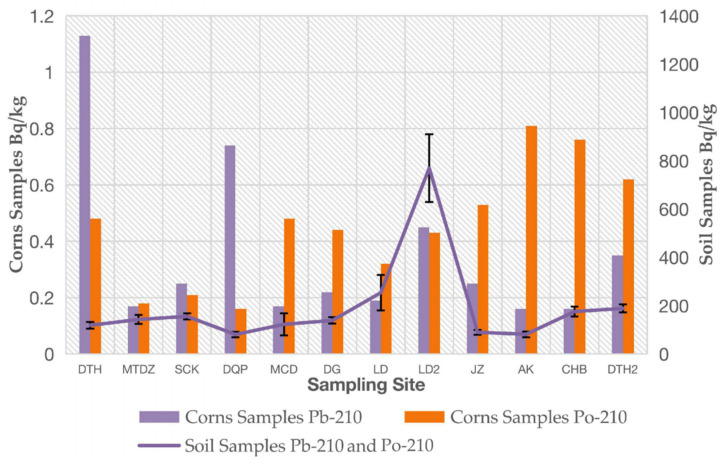
The activity concentrations of ^210^Pb and ^210^Po in corn samples and their corresponding farmland soil samples.

**Figure 4 molecules-27-02112-f004:**
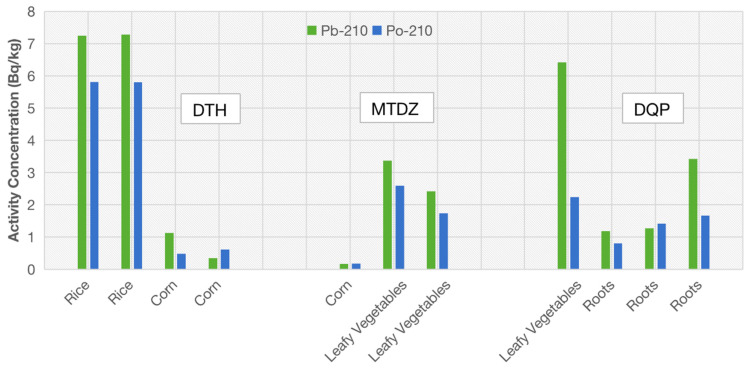
The radionuclide concentrations in different types of crops in 3 villages.

**Figure 5 molecules-27-02112-f005:**
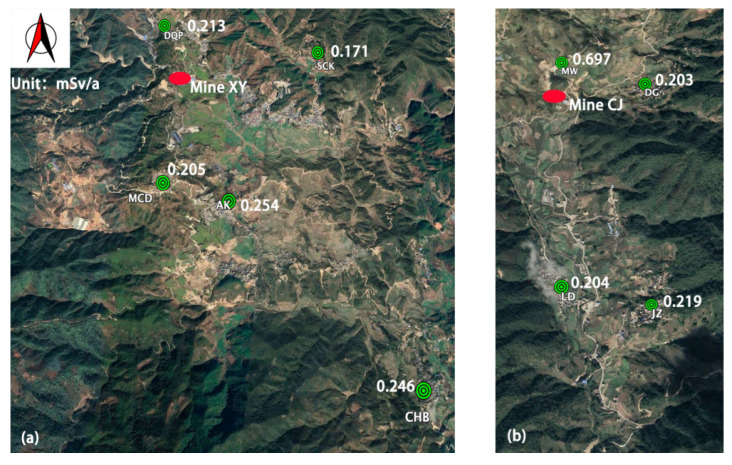
The ingestion doses estimated for adult residents living in the research area (**a**) villages around Mine XY; (**b**) villages around Mine CJ.

**Table 1 molecules-27-02112-t001:** Dose coefficient (DF) for ingestion dose (Sv/Bq).

Radionuclides	DF
^210^Pb	6.9 × 10^−7^
^210^Po	1.2 × 10^−6^

**Table 2 molecules-27-02112-t002:** Radioactivity concentrations in coal ores and residues (Bq/g).

Samples	Points	^238^U	^232^Th	^226^Ra	^40^K
Coal ores	5	0.06–1.82	0.04–0.28	0.01–1.65	0.15–0.60
Residues	5	0.19–3.19	0.07–0.26	0.18–3.29	0.58–1.06

**Table 3 molecules-27-02112-t003:** Radioactivity concentrations in fly ashes and chemical residues (Bq/g).

Samples	Points	^238^U	^226^Ra	^210^Pb	^210^Po	^232^Th
Fly ashes	1	4.80	4.73	42.9	24.3	0.13
Chemical residues	3	2.10–2.40	0.07–5.30	12.5–25.3	17.9	0.01

**Table 4 molecules-27-02112-t004:** Radioactivity concentrations in rivers.

Samples	Points	Total Th (μg/L)	Total U (μg/L)	^226^Ra (mBq/L)	^210^Po (mBq/L)	^210^Pb (mBq/L)
Contaminated water	6	0.93–2.64	1.68–6.78	11.5–68.2	1.40–7.93	49.2–427.7
Background	1	0.44	0.28	3.08	7.78	1.26

**Table 5 molecules-27-02112-t005:** Activity concentrations of ^210^Pb and ^210^Po in samples (Bq/kg·DW).

Samples	Numbers	^210^Pb Average	Range	^210^Po Average	Range	Types
Rice	2	7.26	7.25–7.27	5.81	5.80–5.81	Cereals
Wheat	1	1.92	1.92	2.73	2.73	Cereals
Corn	12	0.35	0.16–1.13	0.45	0.16–0.81	Cereals
Greens	3	4.07	2.42–6.42	2.19	1.74–2.59	Leafy vegetables
Plantains	3	1.63	1.19–3.42	1.32	0.80–1.67	Root vegetables

**Table 6 molecules-27-02112-t006:** Representative adults consumption rates among local residents.

**Vegetative** **Food**	**Cereals**	**Root** **Vegetables**	**Leafy** **Vegetables**
	**(kg·DW/a)**	**(kg·FW/a)**	**(kg·FW/a)**
	126.2	29.1	132.3

**Table 7 molecules-27-02112-t007:** The annual ingestion doses (mSv/a) of adults due to the intake of ^210^Pb and ^210^Po via the consumption of different vegetables.

Villages	Cereal		Root Vegetable		Leafy Vegetable		Total Dose
	^210^Pb	^210^Po	^210^Pb	^210^Po	^210^Pb	^210^Po	
MW	0.167	0.412	0.010	0.013	0.059	0.036	0.697
DTH	0.350	0.480	0.010	0.013	0.059	0.036	0.948
SCK	0.022	0.032	0.010	0.013	0.059	0.036	0.171
DQP	0.064	0.024	0.013	0.017	0.059	0.036	0.213
MCD	0.015	0.072	0.010	0.013	0.059	0.036	0.205
DG	0.019	0.066	0.010	0.013	0.059	0.036	0.203
LD	0.030	0.057	0.010	0.013	0.059	0.036	0.204
JZ	0.022	0.080	0.010	0.013	0.059	0.036	0.219
AK	0.014	0.122	0.010	0.013	0.059	0.036	0.254
CHB	0.014	0.115	0.010	0.013	0.059	0.036	0.246
Average	0.067	0.135	0.010	0.014	0.056	0.035	0.336
MTDZ	0.015	0.027	0.010	0.013	0.027	0.033	0.125

**Table 8 molecules-27-02112-t008:** The annual ingestion doses contributed by ^210^Pb and ^210^Po in foods (mSv/a).

Reference Level	Cereal		Leafy Vegetables		Root Vegetables		Total Intake Dose
	^210^Pb	^210^Po	^210^Pb	^210^Po	^210^Pb	^210^Po	mSv/a
	Bq/kg	Bq/kg	Bq/kg	Bq/kg	Bq/kg	Bq/kg	
World	0.05	0.06	0.08	0.10	0.03	0.04	0.042
China	0.03	0.04	0.36	0.43	0.03	0.03	0.112

For the world reference level here, the food consumption rate of cereals, root vegetables and leafy vegetables for adults is 287.6 kg/a was the same food consumption rate in the research area used in the dose calculation.

## Data Availability

The data presented in this study are available in the article.

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
