# Peer review of "Estimation of the Annual Effective Dose Due to the Ingestion of 210Pb and 210Po in Crops from a Site of Coal Mining and Processing in Southwest China"

_molecules, 2022, doi:10.3390/molecules27072112_

Round 1

Reviewer 1 Report

This study estimated concentrations of 210Pb and 210Po in crops around mining sites and ingestion dose due to these radionuclides. It presents a significant addition to data on natural radionuclide concentrations and resulting doses. Some minor comments are shown below.

Page 1, Line 6 of Introduction section

References should be numbered in an order which they appear, I think. Reference No. 21 should be numbered as No.4.

The same comment will be given for Reference No.30 (cited in the second paragraph of Page 2).

Section 2.3 Analysis Methods, first line

“Corns, wheats, rice, greens and plantains grown in the research area were collected as representative samples.”

The number of samples for each crop is shown in Table 5. Only one sample was taken for wheats. How did you consider this sample as “representative”? If it’s difficult to explain, it would be better to revise the above sentence.

3.1 Monitoring Data, first line

“Samples of raw materials, products and waste residues were collected…”

Methods for collecting these samples and their analysis are not shown in Materials and Methods chapter, I think. At least, information on sampling points and number of samples should be shown.

(Under Table 3)

“Radioactivity concentrations in water collected in research area…”

The same as above. It would be better to give information on sampling points and number of samples.

Table 4 “uncontaminated water”

According to the last line of section 2.1 “Site Description”, “samples from Village MTDZ were considered as a reference level, or the background level of this area”. Do you mean water sampled from Village MTDZ as “uncontaminated water”?

Table 7

The dose due to leafy vegetable intake is the same for all villages, while the dose due to cereal intake is different among the villages. How were these calculated? According to Table 5, three samples were taken for leafy vegetables. Did you use an average concentration for calculating ingestion dose due to intake of leafy vegetable for all villages?

Also, the dose due to root vegetable intake is the same except one village (DQP). I’m wondering how it was calculated. Some explanation will be needed.

References

Some of the authors in the reference list are written in Initials only.

Reviewer 2 Report

Dear authors

I recommend the reading of my reviewer report which is attached in this form. In that document I address to you my criticism and suggestions to make this manuscript ready for editing.

Whish you well

Reviewer 3 Report

The authors investigated the Annual Effective Dose Due to Ingestion of 210Pb and 210Po in Crops from A Site of Coal Mining and Processing in Southwest China

Abstract: Was a very good summary of the results and conclusions.
Introduction: The background studies done prior to this one were well identified and the gap in them too. Relevant literature was given.

Problem statement: Authors identified the major problem in this Cola mine area. Viz: The farmlands around the site of coal mining and germanium processing have been contaminated by the solid waste and mine water.

Methodology: Although the authors use PIPS Alpha spectrometer, the use of Polonium tracer provided quality assurance to the results. (Liquid Scintillation counting will have been better.

Results and discussions: From the analysis, the results show that crops grown on contaminated farmland contained an enhanced level of radioactivity concentration, which was apparently higher than the China average level of 0.112 mSv/a, and the world average level of 0.042 mSv/a through 210Pb and
210Po in crops intake respectively. This showed that the ingestion of crops from these farmlands is not safe for farm dwellers.

Author Response

We appreciate your comments and suggestions a lot!

This manuscript is a resubmission of an earlier submission. The following is a list of the peer review reports and author responses from that submission.